# Heterozygous Variants in *FREM2* Are Associated with Mesiodens, Supernumerary Teeth, Oral Exostoses, and Odontomas

**DOI:** 10.3390/diagnostics13071214

**Published:** 2023-03-23

**Authors:** Piranit Nik Kantaputra, Kanich Tripuwabhrut, Robert P. Anthonappa, Kanoknart Chintakanon, Chumpol Ngamphiw, Ploy Adisornkanj, Nop Porntrakulseree, Bjorn Olsen, Worrachet Intachai, Raoul C. Hennekam, Alexandre R. Vieira, Sissades Tongsima

**Affiliations:** 1Center of Excellence in Medical Genetics Research, Faculty of Dentistry, Chiang Mai University, Chiang Mai 50200, Thailand; 2Division of Pediatric Dentistry, Department of Orthodontics and Pediatric Dentistry, Faculty of Dentistry, Chiang Mai University, Chiang Mai 50200, Thailand; 3Division of Orthodontics, Department of Orthodontics and Pediatric Dentistry, Faculty of Dentistry, Chiang Mai University, Chiang Mai 50200, Thailand; 4Department of Pediatric Dentistry, University of Western Australia Dental School, Nedlands, WA 6009, Australia; 5National Biobank of Thailand, National Science and Technology Development Agency, Khlong Luang 12120, Thailand; 6Dental Department, Sawang Daen Din Crown Prince Hospital, Sakon Nakhon 47110, Thailand; 7Dental Department, Lumphun Hospital, Lumphum 51000, Thailand; 8Department of Developmental Biology, Harvard School of Dental Medicine, Harvard University, Boston, MA 02115, USA; 9Department of Pediatrics, Academic Medical Center, University of Amsterdam, 1105 AZ Amsterdam, The Netherlands; 10Department of Oral and Craniofacial Sciences, School of Dental Medicine, University of Pittsburgh, Pittsburgh, PA 15213, USA

**Keywords:** dental anomalies, extra teeth, Fraser syndrome, supernumerary tooth, tooth formation, mesiodens, mesiodentes

## Abstract

Background: Supernumerary teeth refer to extra teeth that exceed the usual number of dentitions. A mesiodens is a particular form of supernumerary tooth, which is located in the premaxilla region. The objective of the study was to investigate the genetic etiology of extra tooth phenotypes, including mesiodens and isolated supernumerary teeth. Methods: Oral and radiographic examinations and whole-exome sequencing were performed on every patient in our cohort of 122 patients, including 27 patients with isolated supernumerary teeth and 94 patients with mesiodens. A patient who had multiple supernumerary teeth also had odontomas. Results: We identified a novel (c.8498A>G; p.Asn2833Ser) and six recurrent (c.1603C>T; p.Arg535Cys, c.5852G>A; p.Arg1951His, c.6949A>T; p.Thr2317Ser; c.1549G>A; p.Val517Met, c.1921A>G; p.Thr641Ala, and c.850G>C; p.Val284Leu) heterozygous missense variants in *FREM2* in eight patients with extra tooth phenotypes. Conclusions: Biallelic variants in *FREM2* are implicated in autosomal recessive Fraser syndrome with or without dental anomalies. Here, we report for the first time that heterozygous carriers of *FREM2* variants have phenotypes including oral exostoses, mesiodens, and isolated supernumerary teeth.

## 1. Introduction

Supernumerary teeth refer to extra teeth that exceed the usual number in dentitions, which either erupt or remain unerupted [1,2,3]. Supernumerary teeth are rare in primary dentition [4]. The prevalence of supernumerary teeth has been reported to be 0.04–3.8%, depending on the studied population and the method of selecting the samples [2,3,5,6,7]. The prevalence of supernumerary teeth is higher in Asian populations [5,8,9]. Several supernumerary teeth are unerupted. Therefore, the reported low prevalence (0.04%) of supernumerary teeth might be because radiography is not used [6]. They are more common in males than they are in females (M:F; 2.2:1). Supernumerary teeth may occur as a single tooth or multiple teeth in any region of the jaws in the same person. Approximately 12–23% of supernumerary teeth are bilateral [3,5].

The upregulation of WNT/β-catenin and SHH signaling is implicated in supernumerary tooth formation [7,10]. Supernumerary teeth are associated with malformation syndromes, including *APC*-associated familial adenomatous polyposis, *RUNX2*-associated cleidocranial dysplasia [11], *TRPS1*-associated tricho-rhino-phalangeal syndrome [10,12,13,14], *ROR2*-associated Robinow syndrome, and *CREBBP*-associated Rubinstein–Taybi syndrome [11]. Regarding genes for non-syndromic supernumerary teeth, only a variant in *PDGFRB* and a variant in *FER1L6* have been reported to be associated with isolated or non-syndromic supernumerary teeth [15,16].

Mesiodens, the most common type of supernumerary tooth, is a particular type of supernumerary tooth that is found in the premaxilla region. The prevalence of mesiodens is about 2%, depending on the studied population and the methods of including patients [17,18,19,20]. Regarding familial cases of mesiodens, only mesiodens are seen in the family, and they do not occur with other supernumerary teeth. Mesiodens is inherited as autosomal dominant inheritance, with about 50% penetrance. Mesiodens and supernumerary teeth generally are not present in the same persons. A large number of mesiodens are inverted or unerupted [21,22,23,24,25]. Mesiodens is a genetically heterogeneous dental malformation. Recently, genetic variants in *LOW DENSITY LIPOPROTEIN RECEPTOR-RELATED PROTEIN 5* (*LRP5;* MIM 603506), *LOW DENSITY LIPOPROTEIN RECEPTOR-RELATED PROTEIN 6* (*LRP6*; MIM 603507), *WNT LIGAND SECRETION MEDIATOR* (WLS; MIM 611514), *DICKKOPF WNT SIGNALING PATHWAY INHIBITOR 1* (DKK1; MIM 605189), and *LOW DENSITY LIPOPROTEIN RECEPTOR-RELATED PROTEIN 4* (LRP4; MIM 604270) have been reported to be implicated in mesiodens with or without oral exostoses [21,22,23,24,25].

*FRAS1*-related extracellular matrix protein 2 (FREM2; MIM 608945) is a member of the FREM2-FRAS1-FREM1 complex, which contributes to epithelial–mesenchymal coupling [26]. Biallelic variants in *FREM2, FREM1*, or *FRAS1* result in Fraser syndrome (MIM 607830; 608945; 604597), a rare autosomal recessive malformation syndrome characterized by cryptophthalmos, syndactyly, urogenital abnormalities, and dental anomalies. The phenotypes of Fraser syndrome caused by variants in *FREM2, FREM1*, or *FRAS1* are not distinguishable [27,28].

Abnormalities of epithelial–mesenchymal interactions during tooth formation are implicated in dental malformations [29]. Fraser syndrome patients with *FRAS1* or *FREM2* variants share dental findings, including tooth agenesis and shortened roots, suggesting that the dental anomalies result from an impaired FRAS1-FREM2-FREM1 complex and subsequent abnormal epithelial–mesenchymal interactions. Carriers of *FREM2* heterozygous variants including the heterozygous parents of patients with FRASER syndrome have not been reported to have dental anomalies [27,28].

We investigated 122 patients with extra tooth phenotypes and found a novel and six heterozygous missense *FREM2* variants in eight patients affected by mesiodens, supernumerary teeth, or supernumerary teeth with odontomas. This is the first report describing that heterozygous *FREM2* variant carriers carry phenotypes that are extra tooth phenotypes. The fact that heterozygous *FREM2* was thought to not being affected might be due to the absence of evaluations of dental development.

## 2. Materials and Methods

### 2.1. Patient Recruitment

The Human Experimentation Committee of the Faculty of Dentistry, Chiang Mai University (no. 71/2020), approved this study. It was performed following the ethical standards of the 1964 Declaration of Helsinki and its later amendments or comparable ethical standards. Informed consent was obtained from participants or their parents.

### 2.2. Patients

Oral and radiographic examinations were performed on our cohort of 122 patients affected by various kinds of isolated extra tooth phenotypes, which included 94 patients with isolated mesiodens and 27 patients (10 females; 17 males) with isolated supernumerary teeth. Regarding 94 patients with mesiodens, 64 (68.1%) were males and 30 (31.9%) were females. Seventy-eight patients (82.9%) had single mesiodens, while 16 (17.1%) of them had double mesiodentes. The orientation of the mesiodens was noted in 64 mesiodentes of 54 patients; 43 (67.1%) of them had a normal orientation, 20 (31.3%) were inverted ones, and 1 (1.6%) had a transverse orientation. The eruption status was noted in 60 mesiodentes of 49 patients: 32 (53.3%) were erupted and 28 (46.7%) were unerupted. Regarding the patients with isolated supernumerary teeth that were not mesiodens, the information of the supernumerary teeth was not available. A patient with supernumerary teeth also was found to have odontomas. Inclusion criteria of the study were patients with isolated extra tooth phenotypes, including supernumerary teeth or mesiodens. Exclusion criteria of the study were patients with the usual number of or fewer teeth or syndromic supernumerary teeth.

### 2.3. Whole-Exome Sequencing, Mutation Analysis, and Bioinformatic Analyses

The genomic DNA of all patients was isolated from saliva using the Oragene-DNA (OG-500) Kit (DNA Genotek, CANADA). One hundred and twenty-two patients with isolated mesiodens or isolated supernumerary teeth underwent the whole-exome sequencing (WES) service from Macrogen Inc., Seoul, Korea. The WES service employed SureSelect V6 kit (PR7000-0152; Agilent Technologies, Santa Clara, CA, USA), including untranslated regions, resulting in a 80× read depth on average. We used GATK3.8 best practices to identify variants, while BWA-mem was used to map the raw sequencing reads (FASTQ) with the GRCh38+decoy human genome reference sequence. In order to forecast deleterious mutational effects, the reported variants from GATK3.8 were loaded into the Variant Effect Predictor (VEP) with the plugin Database of Nonsynonymous Functional Prediction (dbNSFP), which is a mutation prediction database containing >30 variants’ impact prediction algorithms. The predicted variants of every patient were reported in a variant calling file (VCF file). We collected these files and stored them in our in-house database, which allowed us to query pathogenic variants according to different segregation modes. In addition, variant allele frequencies were determined by comparing them against well-known public variant databases, including gnomAD (https://gnomad.broadinstitute.org (accessed on 8 January 2023), 1000 Genomes (https://www.genome.gov/27528684/1000-genomes-project (accessed on 8 January 2023), GenomeAsia (https://browser.genomeasia100k.org), and the Thai Reference Exome database (T-Rex; https://genomicsthailand.com/Search/T-Rex (accessed on 8 January 2023). The following factors were taken into account when we were prioritizing the variants: (1) the rarity of the allele; (2) the CADD score > 15; (3) localization to or near an essential functional region of the protein. Then, if familial samples were available, the validated variants of interest were examined for segregation using PCR-based amplification and Sanger sequencing. The effects of mutations were predicted by MutationTaster (https://www.mutationtaster.org (accessed on 8 January 2023)). PolyPhen-2 (http://genetics.bwh.harvard.edu/pph2/ (accessed on 8 January 2023)), and SIFT (https://sift.bii.a-star.edu.sg (accessed on 8 January 2023)).

## 3. Results

We identified a novel (c.8498A>G; p.Asn2833Ser) and six heterozygous missense variants (c.1603C>T; p.Arg535Cys (rs201457616), c.5852G>A; p.Arg1951His (rs201806885), c.6949A>T; p.Thr2317Ser, c.1549G>A; p.Val517Met (rs566143955), c.1921A>G; p.Thr641Ala (rs116802472), and c.850G>C; p.Val284Leu (rs770004356) in *FREM2* in five patients with mesiodens, two patients with supernumerary teeth, and a patient with supernumerary teeth and odontomas (Table 1; Figure 1, Figure 2, Figure 3, Figure 4 and Figure 5). Patient 1 had torus palatinus (Figure 2A). Patient 7 had torus palatinus and torus mandibularis (Figure 5A,B).

The heterozygous missense variant c.1603C>T; p.Arg535Cys was identified in patient 1, who had a mesiodens and torus palatinus. According to gnomAD, this variant is rare, with a global allele frequency of 0.0001712. It has not been reported in the South Asian population. Its allele frequency in the East Asian population is 0.002164. According to the T-Rex database, this allele was found in 4 out of 2184 alleles (allele frequency = 0.0018315 or 0.18%). The allele frequency of this variant in the study group (0.409836%) is higher than those in the normal Thai population (T-Rex database) and in gnomAD (Table 2). This variant is predicted to be disease causing, possibly damaging, and damaging by MutationTaster (prob = 0.99999999580175), PolyPhen-2 (0.994), and SIFT (0.001), respectively. The CADD and DANN scores of this variant are 25.4 and 0.9992, respectively (Table 1).

The heterozygous missense variant c.5852G>A; p.Arg1951His was identified in patient 2, who had erupted double mesiodens. This variant is extremely rare, with a global allele frequency of 0.000003982 according to gnomAD. It has not been reported in the South Asian population. Its allele frequency in the East Asian population is 0.00005448. This variant has not been reported in the normal Thai population (T-Rex database). The allele frequency of this variant in the study group (0.409836%) is higher than those in the T-Rex and gnomAD databases (Table 2). This variant is predicted to be disease causing (prob = 0.99712280816916) and possibly damaging (0.873) by MutationTaster and PolyPhen-2, respectively. The CADD and DANN scores of this variant are 19.18 and 0.9985, respectively (Table 1).

Patient 3 had double mesiodens; one had erupted and the other had not. He carried a heterozygous missense variant, c.6949A>T; p.Thr2317Ser, in the *FREM2* gene. This variant cannot be found in e, LOVD, and HGMD databases (Table 1), but according to the T-Rex database, it is reported in one of two thousand, one hundred, and eighty-four alleles. The allele frequency of this variant in the study group (0.409836%) is higher than those in the normal Thai population (T-Rex database) and gnomAD (Table 2). This variant is predicted to be disease causing (prob: 0.999998598060573) by MutationTaster. The CADD and DANN scores of this variant are 22.4 and 0.9985, respectively (Table 1).

Patient 4, who had an inverted and unerupted double mesiodens, carried a heterozygous missense variant, c.1549G>A; p.Val517Met, in the *FREM2* gene. This variant is very rare, with a global allele frequency of 0.00002012 according to gnomAD. This variant has not been reported in the South Asian population. The allele frequency of this variant in the East Asian population is 0.0002730. This allele is not reported in T-Rex database. The allele frequency of this variant in the study group (0.409836%) is higher than those in T-Rex and gnomAD databases (Table 2). This variant is predicted to be disease causing (0.999999999751131), possibly damaging (1.000), damaging (0) by MutationTaster, Poly-Phen-2, and SIFT, respectively. The CADD and DANN scores are 26.3 and 0.0001, respectively.

The heterozygous missense variant c.1921A>G; p.Thr641Ala was identified in patient 5, who had two mandibular supernumerary teeth and taurodontism, and patient 6, who had a mesiodens, agenesis of the maxillary permanent third molars, and taurodontism of the maxillary left permanent second molar. According to gnomAD, this variant is very rare, with a global allele frequency of 0.00002829. The allele frequency of this variant in the East Asian population is 0.0003510. It has not been reported in the South Asian population. According to the T-Rex database, this allele was found in 5 out of 2184 alleles (allele frequency = 0.00228938 or 0.22%). The allele frequency of this variant in the study group (0.409836%) is higher than those in T-Rex and gnomAD databases (Table 2). This variant is predicted to be disease causing (prob: 0.999961580268265), possibly damaging (0.474), and damaging (0.008) by MutationTaster, PolyPhen-2, and SIFT, respectively. The CADD and DANN scores of this variant are 22.1 and 0.9889, respectively. In addition to the *FREM2* variant, patient 5 also had a heterozygous missense variant (c.136A>T; p.Met46Leu; chr1 g.68659881T>A; rs368633951) in *WLS*, and this has been reported [23]. This allele is rare, with a global allele frequency of 0.0000398.

Patient 7 who had two supernumerary mandibular premolars, unseparated roots of the permanent molars, torus palatinus, and torus mandibularis and carried a heterozygous missense variant, c.850G>C; p.Val284Leu, in the *FREM2* gene. This variant is rare with a global allele frequency of 0.0002231 according to gnomAD. The allele frequency of this variant in the East Asian population is 0.003158. This variant has not been reported in the South Asian population. According to the T-Rex database, this allele is found in 4 out of 2184 alleles (allele frequency = 0.0018315 or 0.18%). The allele frequency of this variant in the study group (0.409836%) is higher than those in the T-Rex and gnomAD databases (Table 2). This variant is predicted to be disease causing (prob: 0.999963783102466) by MutationTaster. The CADD and DANN scores of this variant are 23.8 and 0.9953, respectively.

Patient 8, who had multiple supernumerary teeth and multiple odontomas, carried a heterozygous missense variant, c.8498A>G; p.Asn2833Ser, in the *FREM2* gene (Figure 6). This variant was not found in the T-Rex, gnomAD, LOVD, and HGMD databases; therefore, it is considered to be novel. This mutation is predicted to be disease causing (0.999943878525573) and pathogenic (0.9953) by MutationTaster. The CADD and DANN scores of this variant are 17.14 and 0.9802, respectively (Table 1 and Table 2).

## 4. Discussion

Seven heterozygous missense variants in *FREM2* were identified in eight Thai patients affected with extra tooth phenotypes, including mesiodens, supernumerary teeth, and supernumerary teeth with odontomas (Table 1). The c.1603C>T; p.Arg535Cys (rs201457616), c.5852G>A; p.Arg1951His (rs201806885), c.1549G>A; p.Val517Met (rs566143955), c.1921A>G; p.Thr641Ala (rs116802472), and c.850G>C; p.Val284Leu (rs770004356) variants are rare, and according to gnomAD, their global allele frequencies are 0.0001712, 0.000003982, 0.00002012, 0.00002829, and 0.0002231, respectively. The c.6949A>T; p.Thr2317Ser variant in *FREM2* is not reported in gnomAD, LOVD, and dbSNP, but an allele was found in one out of two thousand, one hundred, and eighty-four person alleles (allele frequency = 0.00045788) in the T-Rex database (Table 1).

### 4.1. FREM2 Variants and Their Pathogenicities

None of the seven variants have been reported before as being pathogenic. The variant c.8498A>G p.Asn2833Ser is not reported in gnomAD, LOVD, and NCBI and is considered to be novel. The finding of the c.1921A>G; p.Thr641Ala variant in two unrelated patients, one (patient 6) with mesiodens, and the other one (patient 5) with two supernumerary mandibular premolars and taurodontism supports its pathogenicity. The allele frequencies of the variants found in our patients are higher than those in the normal Thai (T-Rex), South Asian, East Asian, and global populations, according to gnomAD (Table 2).

For rare variants as the causes of disease, statistical analyses are not appropriate as cases due to them being rare, and thus, not sufficient. It is only possible for more common conditions. As it is, the phenotypic presentation of supernumerary tooth phenotypes is likely to be under-recognized in the general population, and especially, an isolated feature is not likely to be considered as a pathology warranting exclusion or mention in public genomic databases such as gnomAD (considered ‘healthy’ individuals) from which variant frequencies that underpin such causal relationships are made.

With respect to ‘control’ populations, it is necessary to note that the prevalence of supernumerary teeth has been reported to be 0.04–3.8% [2,3,5,6,7], but most supernumerary teeth including mesiodens have not erupted into the oral cavities. Notably, more than 50% of mesiodens are unerupted [21,22,23,24,25]. Therefore, unless all those healthy people included in gnomAD have had radiography-based oral examinations, it is ideally unsuitable to include them as a control cohort, as hundred or more of them might have supernumerary tooth phenotypes.

Rare genetic variants have been claimed as predisposing factors or causal factors for genetic diseases. We are aware that not all rare or novel variants are associated with genetic diseases. However, we are convinced that the rare *FREM2* variants found in our patients are associated with the supernumerary tooth phenotypes because it is well known that biallelic variants in *FREM2* genes are associated with Fraser syndrome with dental anomalies [27,28].

### 4.2. FREM2 Variants, Probability of Being the Loss-of-Function Intolerant (pLI), and the Clinical Significance

In the present cohort of 122 patients with supernumerary tooth phenotypes, the frequency of *FREM2* variants that are missense variants is 0.05737705 (7/122). An estimate of the pLI can be used to determine whether or not a single disrupting variant is of likely clinical significance [30]. pLI is derived from comparing the number of protein-truncating variants in a sample with the number of expected mutations under neutrality, given an estimated mutation rate for the gene of interest. In the present study’s cohort of individuals with mesiodens and supernumerary teeth, the frequency of missense variants was 0.05737705. Assuming that the number of segregating missense variants observed in a gene is Poisson-distributed, it will correspond to the mean of the number of segregating protein-truncating variants expected in the sample under neutrality times the depletion in the number due to selection. These estimates were categorized as being either neutral (mean = 1), recessive (mean = 0.463), or haploinsufficient (mean = 0.089) (28). The frequency of *FREM2* variants in the present cohort (0.05737705) is very similar to the estimate of missense variants leading to haploinsufficiency, and this agrees with the findings of heterozygous variants only. It is noteworthy that, such as *FREM2*, the *WNT10A* and *WNT10B* genes have pLI scores of 0, and heterozygous variants in *WNT10A* and *WNT10B* are well known to be implicated in dental anomalies [31]. It has been argued that pLI scores are unsuitable for evaluating autosomal recessively acting genes due to the lack of consequences in heterozygotes [32]. Indeed, pLI scores may not be suitable to evaluate variants that cause dental phenotypes, as these are generally mild or may even go unnoticed, and pLI is calculated based on individuals mentioned in gnomAD. Therefore, if the phenotype is not life-threatening or does not result in a severe disability, such as having a supernumerary tooth, the pLI score is not likely to be meaningful.

### 4.3. The Absence of Rare Variants in Other Known Dental Anomaly-Related Genes

In addition to *FREM2* variants, a rare variant (c.130A>T; p.Met44Leu) in the *WLS* gene was found in patient 5 and has been reported [23]. WES also identified rare variants: the c.33dup; p.Lys12Ter variant in *EVC2* in patient 1 and the c.461C>T; p.Pro154Leu variant in *MSX1* in patient 4, which might contribute to the phenotypes found in both patients (Table 3). Otherwise, exome sequencing in our patients did not reveal rare variants in other known, dental anomaly-related genes such as *WNT10A*, *WNT10B*, *PAX9*, *AXIN2*, *LRP4*, *LRP5*, *LRP6*, *GREM2*, *LAMB3*, *TSPEAR*, *TFAP2B*, *PITX2*, *BMP4*, *EDA*, *EDAR*, *EDARADD*, *EVC*, *EVC2*, *CREBBP*, *COL1A2*, *ANTXR1*, *FGF10*, *SMOC2*, *KDF1*, *KREMEN1*, and *DKK1* [21,22,23,24,25]. The present study demonstrates, for the first time, that heterozygous carriers of *FREM2* variants may have isolated supernumerary teeth, mesiodens, and supernumerary teeth with odontomas. Notably, the dental anomalies reported in biallelic patients with *FREM2*-associated Fraser syndrome were tooth agenesis and short roots [27,28]. Since our patients were heterozygous, it is hypothesized that the *FREM2* variants in them might have caused gain-of-function and resulted in supernumerary teeth, mesiodens, and odontomas. Therefore, the nature of the variants might influence the phenotype.

### 4.4. FREM2 Heterozygous Carriers with Phenotypes

Heterozygous carriers of several autosomal recessive disorders, such as thalassemia, may have mild phenotypes. Biallelic variants in *FREM2* are implicated in Fraser syndrome, and heterozygous carriers of the *FREM2* variants have not been reported to have phenotypes. However, a detailed intraoral evaluation is typically not performed in medical settings, and heterozygous parents of the previously reported patients with Fraser syndrome might have had undetected dental anomalies. Therefore, we encourage clinical and radiographic evaluations of parents of children with *FREM2*-related Fraser syndrome for dental anomalies, which would further prove the pathogenicity.

### 4.5. FREM2, Tooth Development, and Supernumerary Tooth Formation

During tooth development, *Frem2* is expressed in the dental epithelium at the cap stage [27]. Of note, nephronectin plays critical roles in *Sox2* expression, and a mutation in *SOX2* was implicated in SOX2 anophthalmia syndrome with supernumerary teeth [33]. Therefore, it is hypothesized that heterozygous missense variants in *FREM2* result in alterations of the FRAS1-FREM2-FREM1 complex, the disruption of the basement membrane assembly of nephronectin [26], the abnormal expression of *SOX2* [33], the overactivation of WNT/β-catenin signaling [34], and the subsequent formation of supernumerary teeth, mesiodens, and odontomas [29,35,36] (Figure 7). Most cases of mesiodens are sporadic. If cases are familial, the mode of inheritance is autosomal dominance with incomplete penetrance [24,37]. This finding implies that many genes are involved in its etiology. Recently, pathogenic variants in *LRP5*, *LRP6*, *WLS*, *DKK1*, and *LRP4* have been implicated in mesiodens or odontomas [21,22,23,24,25]. Our study suggests that *FREM2* variants are likely the “predisposing factors” to the mesiodens phenotype.

### 4.6. FREM2 Variants and Oral Exostoses

Patients 1 and 6 also had torus palatinus or torus mandibularis, respectively. Oral exostoses, including torus palatinus, torus mandibularis, and buccal exostoses, have been reported to be associated with aberrant WNT/β-catenin signaling [21,23,24]. The oral exostoses found in our patients might have been coincidental or caused by the unknown genetic pathways involving *FREM2*.

### 4.7. FREM2 Variant and Odontomas

We report eight patients, five with mesiodens, two with supernumerary teeth, and one with supernumerary teeth and odontomas. Odontomas are benign mixed odontogenic tumors involving all odontogenic tissues (enamel, dentin, and cementum) [2]. Even though supernumerary teeth and odontomas are classified as distinct entities, both share pathologic processes, topographic distribution, and pathologic manifestations [2]. Recently, odontomas have been reported to be associated with a variant in *LRP6* [21]. Notably, odontomas and supernumerary teeth can be found in patients with *APC*-associated familial adenomatous polyposis syndrome. However, patient 8, who had supernumerary teeth and odontomas, did not have a rare variant in the *APC* gene. Therefore, the phenotype of patient 8 was not related to familial adenomatous polyposis syndrome [11].

### 4.8. Clinical Implication

Our study shows that heterozygous variants in *FREM2* are associated with extra tooth phenotypes. If a patient has an *FREM2* pathogenic variant, it is recommended to perform thorough clinical and radiographic examinations before orthodontic treatment. Supernumerary teeth, mesiodens, and oral exostoses may be presented later in life. If we suppose that a patient’s family member has an *FREM2* variant or a history of having supernumerary teeth, mesiodens, or odontomas, in that case, it is recommended to perform thorough clinical and radiographic examinations and genetic testing of the patient prior to orthodontic treatment to rule out the presence of the *FREM2* variant or other supernumerary tooth-associated genetic variants.

The presence of supernumerary teeth or mesiodens may cause a range of dental complications due to dental crowding, eruption failure, midline diastema, rotation or displacement of the adjacent teeth, root resorption, root dilaceration, dentigerous cyst formation, and the root maldevelopment of permanent teeth [38]. It is important to note that extraction is not always the treatment of choice for supernumerary teeth. Several supernumerary teeth are incidental and harmless. Many of them are best left in place and kept under observation, mainly when the normal eruption of the neighboring teeth has occurred and no associated pathology is seen [3].

Our study also shows that patients with heterozygous variants in *FREM2* may have oral exostoses. Torus mandibularis and torus palatinus in patients may affect the design of the removable orthodontic appliances. It is important to note that these oral exostoses will appear after the patient has passed through the teenage years. Therefore, the phenotypes associated with *FREM2* variants may affect the orthodontic treatments.

### 4.9. Future Studies

Since FREM2 protein is a vital component of the FRAS1-FREM2-FREM1 complex and subsequent abnormal epithelial–mesenchymal interactions, the disruption of epithelial–mesenchymal interactions as a result of *FREM2* mutations may have deleterious effects on tooth development including tooth mineralization [39]. It is suggested that structural composition of the supernumerary teeth should be studied in the future.

We report heterozygous carriers of *FREM2* in patients with extra tooth phenotypes. Bi-allelic variants of *FREM2* are implicated in Fraser syndrome, as previously mentioned [27,28]. The study of dental conditions of the heterozygous parents of the patients affected with Fraser syndrome has not been conducted. It is suggested that dental evaluations be performed in the parents of patients with Fraser syndrome.

## 5. Conclusions

We report for the first time that heterozygous carriers of *FREM2* may have phenotypes including oral exostoses, mesiodens, and isolated supernumerary teeth.

## 6. Study Limitations

Our cohort consists of 122 patients with supernumerary tooth phenotypes. However, we only have DNA samples and dental information of the affected patients who came for oral and radiographic examinations. We are aware that it would have been ideal if we had each patient’s family members to study the co-segregation between genotype and phenotype. It would have strengthened the association between the heterozygous variants in *FREM2* and the phenotypes.

Regarding the use of T-Rex and gnomAD databases as “normal” populations, it is ideally unsuitable to include them as control cohorts, as hundred or more patients might have supernumerary tooth phenotypes because generally having supernumerary teeth is not painful, and most patients who have supernumerary teeth do not even know they have them. In addition, more than 50% of mesiodens have not erupted into the oral cavities.

## Figures and Tables

**Figure 1 diagnostics-13-01214-f001:**
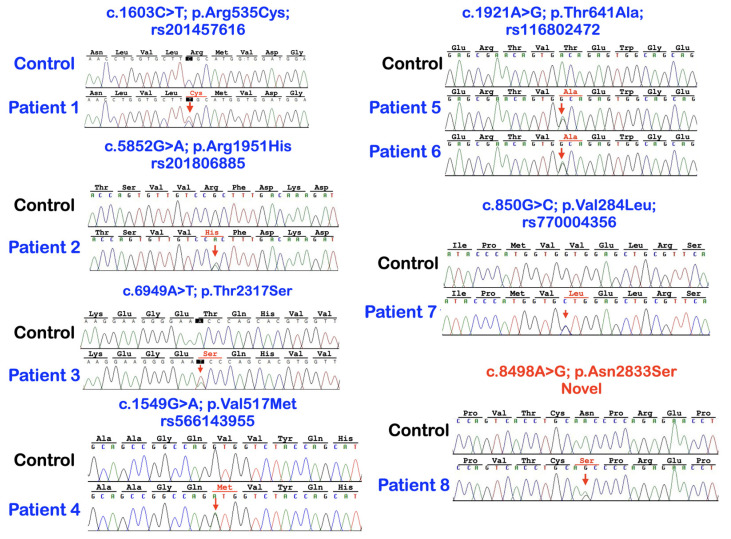
Sequence chromatograms of patients with heterozygous missense variants in *FREM2* gene. Six variants are recurrent and two are novel.

**Figure 2 diagnostics-13-01214-f002:**
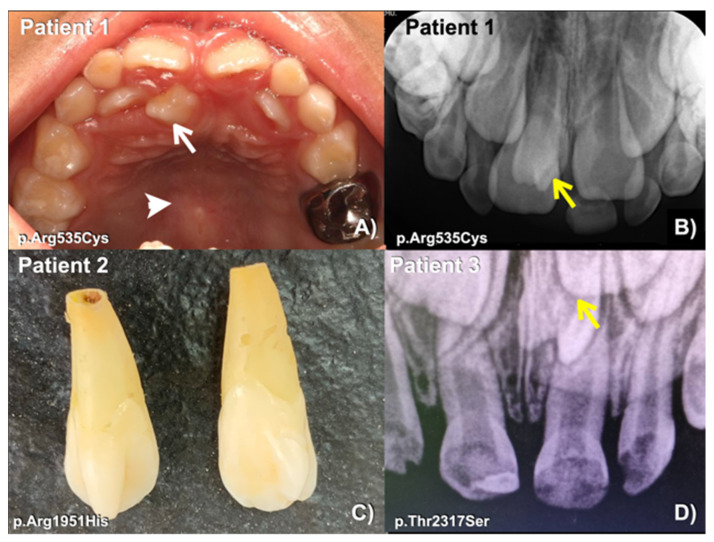
(**A**) Patient 1. Mesiodens (arrow). Torus palatinus (arrowhead). (**B**) Patient 1. Periapical radiograph showing an unerupted mesiodens (arrow). (**C**) Patient 2. Extracted double mesiodens. (**D**) Patient 3. Periapical radiograph showing an unerupted mesiodens (arrow).

**Figure 3 diagnostics-13-01214-f003:**
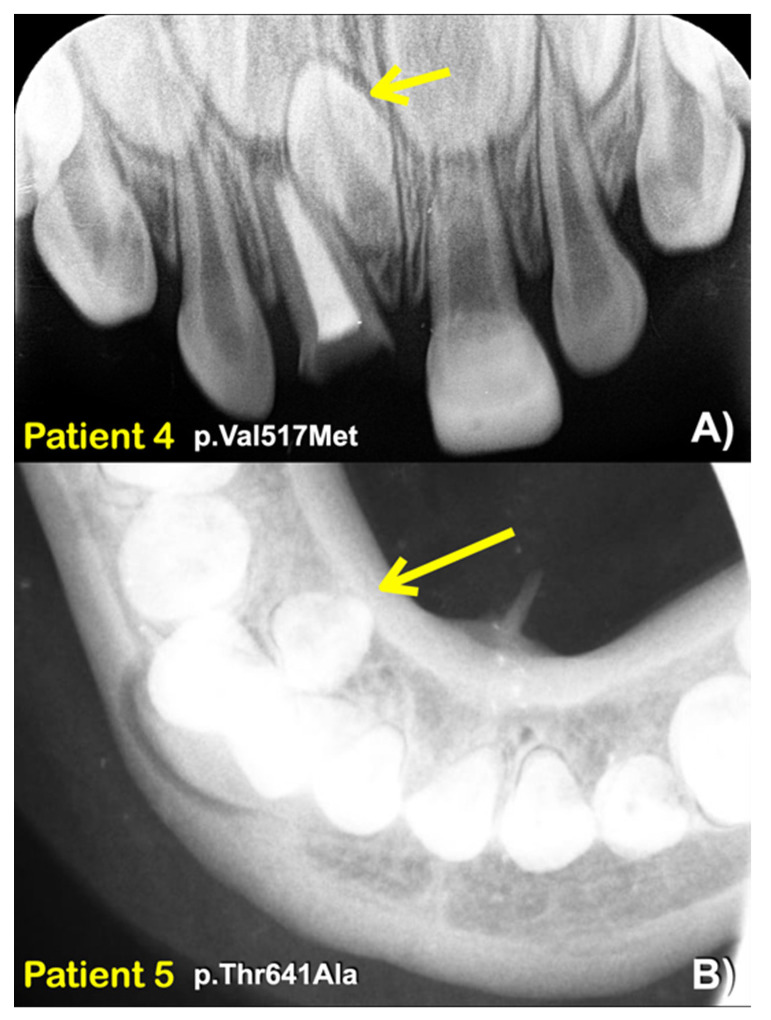
(**A**) Patient 4. A periapical radiograph showing an inverted mesiodens (arrow). (**B**) Patient 5. An occlusal radiograph showing a supernumerary mandibular right premolar (arrow).

**Figure 4 diagnostics-13-01214-f004:**
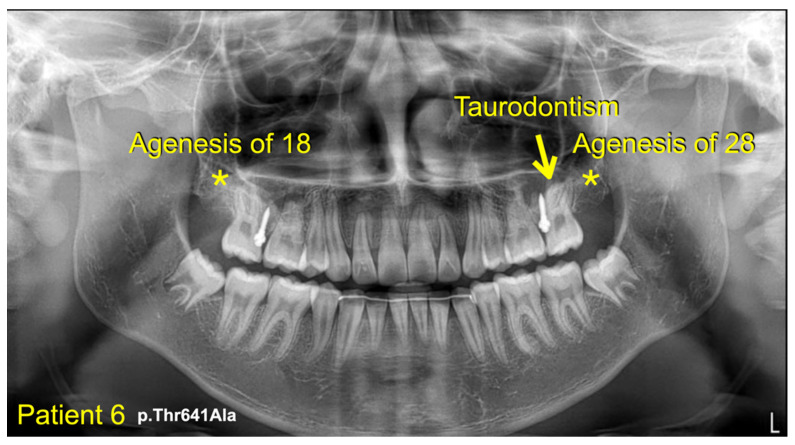
Patient 6. Panoramic radiograph showing agenesis of the maxillary permanent third molars (asterisks). Taurodontism of the left maxillary permanent second molar (arrow).

**Figure 5 diagnostics-13-01214-f005:**
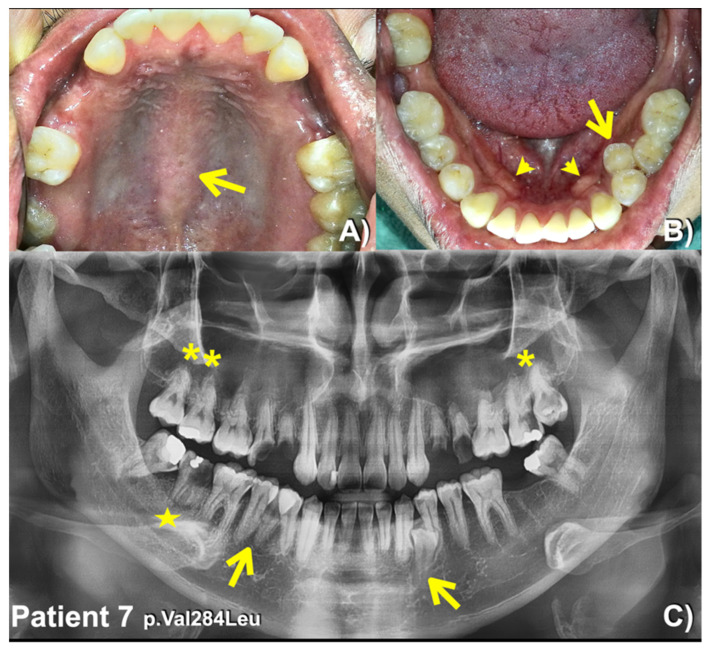
Patient 7. (**A**) Torus palatinus (arrow). (**B**) Torus mandibularis (arrowheads). Supernumerary left mandibular premolar (arrow). (**C**) Panoramic radiograph showing supernumerary mandibular premolars (arrows). Maxillary permanent molars with unseparated roots (asterisks). Short roots of the mandibular right second permanent molar (star).

**Figure 6 diagnostics-13-01214-f006:**
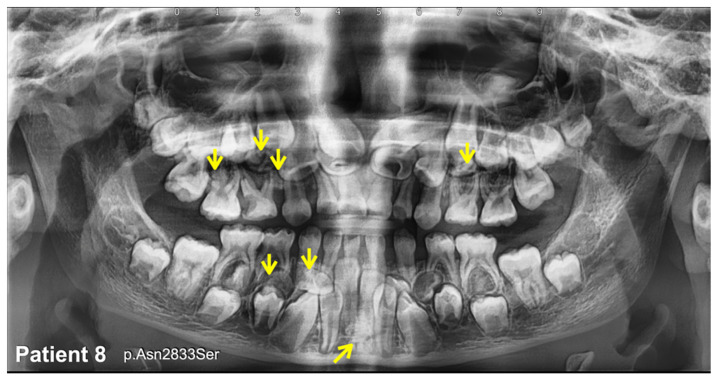
Patient 8. Panoramic radiograph showing multiple supernumerary teeth and multiple odontomas (arrows).

**Figure 7 diagnostics-13-01214-f007:**
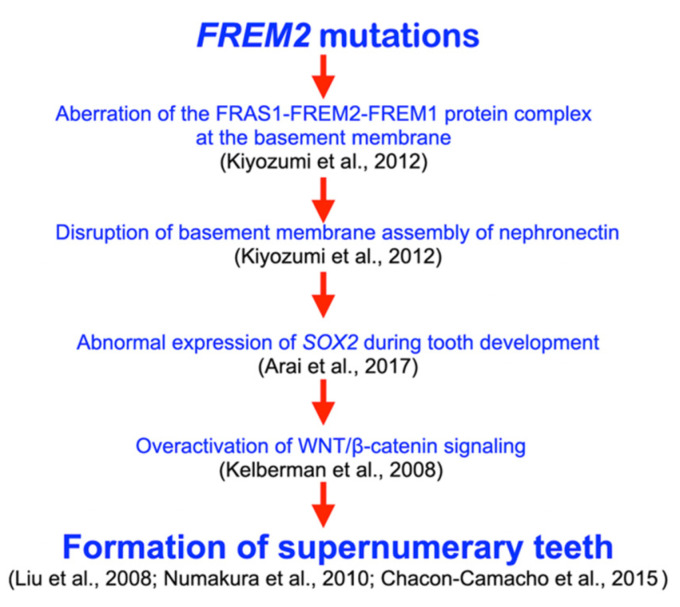
Hypothetical flowchart shows pathogenetic pathway as a result of *FREM2* mutations. Heterozygous missense variants in *FREM2* result in alterations of FRAS1-FREM2-FREM1 complex, disruption of basement membrane assembly of nephronectin, abnormal expression of *SOX2*, overactivation of WNT/β-catenin signaling, and subsequent formation of supernumerary teeth [26,29,33,34,35,36].

**Table 1 diagnostics-13-01214-t001:** Patients with *FREM2* variants, their phenotypes, mutation predictions, and ranking.

Patients	Phenotypes	*FREM2* Variant NM_207361.6; NP_997244.4	Mutation Predictions/Ranking
1 (Female)	Mesiodens and torus palatinus	**c.1603C>T; p.Arg535Cys** rs201457616; chr13-39263084-C-T gnomAD Global = 0.0001712 gnomAD South Asian = 0.000 gnomAD East Asian = 0.002164; dbSNP T = 0.0039 In-house EXOME bank 4/1016 Allele freq in extra tooth cohort 1/244 = 0.409836% Allele freq in T-Rex = 4/2184 (T:0.0018315 or 0.18%)	MutationTaster: disease causing (0.99999999580175) PolyPhen-2: Possibly damaging (0.994) SIFT: Damaging (0.001) CADD: VUS (25.4) DANN: Uncertain (0.9992)
2 (Male)	Mesiodens (double)	**c.5852G>A; p.Arg1951His** rs201806885; chr13-39358778-G-A gnomAD Global = 0.000003982 gnomAD South Asian = 0.000 gnomAD East Asian = 0.00005448 dbSNP A = 0.000 In-house EXOME bank 1/1016 Allele freq in extra tooth cohort 1/244 = 0.409836% Allele freq in T-Rex = 0%	MutationTaster: Disease causing (0.99712280816916) PolyPhen-2: Possibly damaging (0.873) SIFT: TOLERATED (0.068) CADD: Moderate benign (19.18) DANN: Uncertain (0.9985)
3 (Male)	Mesiodens (double; one unerupted and one inverted)	**c.6949A>T; p.Thr2317Ser** chr13-39430286-A-T Not reported in gnomAD and dbSNP In-house EXOME bank 3/1016 Allele freq in extra tooth cohort 1/244 = 0.409836% Allele freq in T-Rex = 1/2184 (T:0.00045788 or 0.04%)	MutationTaster: Disease causing (0.999998598060573) PolyPhen-2: Benign (0.118) SIFT: Tolerated (0.285) CADD: VUS (22.4) DANN: Uncertain (0.9904)
4 (Female)	Mesiodens (single unerupted inverted)	**c.1549G>A; p.Val517Met** rs566143955; chr13-39263030-G-A gnomAD Global = 0.00002012 gnomAD South Asian = 0.000 gnomAD East Asian = 0.0002730 dbSNP A = 0.000 In-house EXOME bank 1/1016 Allele freq in extra tooth cohort 1/244 = 0.409836% Allele freq in T-Rex = 0%	MutationTaster: Disease causing (0.999999999751131) PolyPhen-2: Possibly damaging (1.000) SIFT: Damaging (0) CADD: Pathogenic (26.3) DANN: Uncertain (0.9991)
5 (Female)	Two supernumerary mandibular premolars	**c.1921A>G; p.Thr641Ala** rs116802472; chr13-39263402-A-G gnomAD Global = 0.00002829 gnomAD South Asian = 0.000 gnomAD East Asian = 0.0003510; dbSNP G = 0.000; In-house EXOME bank 3/1016 Allele freq in extra tooth cohort 2/244 = 0.819672% Allele freq in T-Rex = 5/2184 (G:0.00228938 or 0.22%)	MutationTaster: Disease causing (0.999961580268265) PolyPhen-2: POSSIBLY Damaging (0.474) SIFT: Damaging (0.008) CADD: Benign (22.1) DANN: Uncertain (0.9889)
6 (Male)	Mesiodens, agenesis of 18 and 28, and taurodontism of 27		
7 (Male)	Supernumerary mandibular premolars, unseparated roots of molars, torus palatinus, and torus mandibularis	**c.850G>C; p.Val284Leu** rs770004356; chr13-39262331-G-C gnomAD Global = 0.0002231 gnomAD South Asian = 0.000 gnomAD East Asian = 0.003158; dbSNP C = 0.000 In-house EXOME bank 3/1016 Allele freq in extra tooth cohort 1/244 = 0.409836% Allele freq in T-Rex = 4/2184 (G:0.0018315 or 0.18%)	MutationTaster: Disease causing (0.999963783102466) PolyPhen-2: Benign (0.108) SIFT: Tolerated (0.17) CADD: VUS (23.8) DANN: Uncertain (0.9953)
8 (Female)	Multiple supernumerary teeth and multiple odontomas	**c.8498A>G; p.Asn2833Ser (NOVEL)** chr13-39450473-A-G Not reported in gnomAD and dbSNP In-house EXOME bank 2/1016 Allele freq in extra tooth cohort 1/244 = 0.409836% Allele freq in T-Rex = 0%	MutationTaster: Disease causing (0.999943878525573) PolyPhen-2: Benign (0.002) SIFT: Tolerated (0.269) CADD: Moderate benign (17.14) DANN: Uncertain (0.9802)

**Table 2 diagnostics-13-01214-t002:** Allele frequencies of the variants in different populations. Note the frequencies of the *FREM2* variants are higher in the patient cohort than they are in various normal populations.

Patients/ *FREM2* Variants	Patient 1 p.Arg535Cys rs201457616	Patient 2 p.Arg1951His rs201806885	Patient 3 p.Thr2317Ser	Patient 4 p.Val517Met rs566143955	Patient 5 p.Thr641Ala rs116802472	Patient 6 p.Val284Leu rs770004356	Patient 6 p.Asn2833Ser
Frequencies in the study group (*n* = 122 persons or 244 alleles)	N = 1 (0.409836%)	N = 1 (0.409836%)	N = 1 (0.409836%)	N = 1 (0.409836%)	N = 2 (0.819672%)	N = 1 (0.409836%)	N = 1 (0.409836%)
Frequencies in normal Thai population (T-Rex) (*n* = 2184 alleles)	N = 4 (0.18315%)	0%	0%	0%	N = 5 (0.228938%)	N = 4 (0.18315%)	0%
Frequencies in global population gnomAD	0.01712%	0.0003982%	Not reported or 0%	0.0002012%	0.002829%	0.02231%	Not reported or 0%
Frequencies in South Asian population gnomAD	0%	0%	Not reported or 0%	0%	0%	0%	Not reported or 0%
Frequencies in East Asian population gnomAD	0.2164%	0.005448%	Not reported or 0%	0%	0.03510%	0.3158%	Not reported or 0%

**Table 3 diagnostics-13-01214-t003:** Genetic variants with allele frequencies less than 0.0001 (MAF < 0.0001) according to gnomAD in tooth-related genes in patients with *FREM2* variants with mesiodens or isolated supernumerary teeth.

Tooth-Related Genes	Patient 1 3212 Mesiodens	Patient 2 095 Mesiodens (Double)	Patient 3 2952 Mesiodens (Double)	Patient 4 3346 Mesiodens	Patient 5 2775 Supernumerary Premolars	Patient 6 3036 Mesiodens and Tooth Agenesis, and Taurodontism	Patient 7 3099 Supernumerary Premolars, Oral tori, Root Maldevelopment	Patient 8 3262 Multiple Supernumerary Teeth and Multiple Odontomas
*WNT10A*	Not found	Not found	Not found	Not found	Not found	Not found	Not found	Not found
*WNT10B*	Not found	Not found	Not found	Not found	Not found	Not found	Not found	Not found
*PAX9*	Not found	Not found	Not found	Not found	Not found	Not found	Not found	Not found
*AXIN2*	Not found	Not found	Not found	Not found	Not found	Not found	Not found	Not found
*MSX1*	Not found	Not found	Not found	***MSX1* Variant** NM_002448.3:c.461C>T NP_002439.2:p.Pro154Leu rs545651715 chr4:g.4862087C>T MAF = 0.00003574	Not found	Not found	Not found	Not found
*LRP4*	Not found	Not found	Not found	Not found	Not found	Not found	Not found	Not found
*LRP5*	Not found	Not found	Not found	Not found	Not found	Not found	Not found	Not found
*LRP6*	Not found	Not found	Not found	Not found	Not found	Not found	Not found	Not found
*WLS*	Not found	Not found	Not found	Not found	***WLS* Variant** NM_001002292.3:c.130A>T NP_001002292.3:p.Met44Leu chr1:g.68659881T>A rs368633951 MAF: 0.00003891	Not found	Not found	Not found
*DKK1*	Not found	Not found	Not found	Not found	Not found	Not found	Not found	Not found
*BMP4*	Not found	Not found	Not found	Not found	Not found	Not found	Not found	Not found
*GREM2*	Not found	Not found	Not found	Not found	Not found	Not found	Not found	Not found
*TFAP2B*	Not found	Not found	Not found	Not found	Not found	Not found	Not found	Not found
*TSPEAR*	Not found	Not found	Not found	Not found	Not found	Not found	Not found	Not found
*EDA*	Not found	Not found	Not found	Not found	Not found	Not found	Not found	Not found
*EDAR*	Not found		Not found	Not found	Not found	Not found	Not found	Not found
*EDARADD*	Not found	Not found	Not found	Not found	Not found	Not found	Not found	Not found
*PITX2*	Not found	Not found	Not found	Not found	Not found	Not found	Not found	Not found
*EVC*	Not found	Not found	Not found	Not found	Not found	Not found	Not found	Not found
*EVC2*	***EVC2* Variant** NM_001166136.2:c.33dup NP_001159608.1:p.Lys12Ter chr4:g.5699332dup No rs number Not reported in gnomAD	Not found	Not found	Not found	Not found	Not found	Not found	Not found
*COL1A2*	Not found	Not found	Not found	Not found	Not found	Not found	Not found	Not found
*ANTXR1*	Not found	Not found	Not found	Not found	Not found	Not found	Not found	Not found
*FGF10*	Not found	Not found	Not found	Not found	Not found	Not found	Not found	Not found
*SMOC2*	Not found	Not found	Not found	Not found	Not found	Not found	Not found	Not found
*KREMEN1*	Not found	Not found	Not found	Not found	Not found	Not found	Not found	Not found
*KDF1*	Not found	Not found	Not found	Not found	Not found	Not found	Not found	Not found
*ATF1*	Not found	Not found	Not found	Not found	Not found	Not found	Not found	Not found
*DUSP10*	Not found	Not found	Not found	Not found	Not found	Not found	Not found	Not found
*CASC8*	Not found	Not found	Not found	Not found	Not found	Not found	Not found	Not found
*RUNX2*	Not found	Not found	Not found	Not found	Not found	Not found	Not found	Not found
*TRPS1*	Not found	Not found	Not found	Not found	Not found	Not found	Not found	Not found
*C2CD3*	Not found	Not found	Not found	Not found	Not found	Not found	Not found	Not found
*NHS*	Not found	Not found	Not found	Not found	Not found	Not found	Not found	Not found
*MID1*	Not found	Not found	Not found	Not found	Not found	Not found	Not found	Not found
*CREBBP*	Not found	Not found	Not found	Not found	Not found	Not found	Not found	Not found
*EP300*	Not found	Not found	Not found	Not found	Not found	Not found	Not found	Not found
*BCOR*	Not found	Not found	Not found	Not found	Not found	Not found	Not found	Not found
*WNT5A*	Not found	Not found	Not found	Not found	Not found	Not found	Not found	Not found
*DVL1*	Not found	Not found	Not found	Not found	Not found	Not found	Not found	Not found
*DVL3*	Not found	Not found	Not found	Not found	Not found	Not found	Not found	Not found
*IL11RA*	Not found	Not found	Not found	Not found	Not found	Not found	Not found	Not found

## Data Availability

The data underlying this article are available.

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
