# Peer review of "Heterozygous Variants in FREM2 Are Associated with Mesiodens, Supernumerary Teeth, Oral Exostoses, and Odontomas"

_diagnostics, 2023, doi:10.3390/diagnostics13071214_

Round 1

Reviewer 1 Report

The layout of the images could be a little better. Put the clinical photos in one image. Put the content inside the supplementary document in the body of the text.

Author Response

RESPONSE TO COMMENT OF REVIEWER 1

Comment: The layout of the images could be a little better. Put the clinical photos in one image. Put the content inside the supplementary document in the body of the text.

RESPONSE: The authors have moved the content from the supplemental file in the body of the text as suggested. The authors also added Table 2 in the manuscript because the journal missed putting it in at the beginning. Since clinical pictures genuinely describe the important dental phenotypes to the readers especially those who are not dentists, therefore, the authors prefer not to put all clinical pictures in one image.

Thank you for your valuable comments.

Reviewer 2 Report

Manuscript of considerable interest for the dental sector, it needs a major revision before it can be published.

Abstract: to better highlight the results obtained.

Keywords: are they all present on MeSH?

Introduction: all the other pre- and post-natal defects linked to genetic compositions, such as structural anomalies, as already studied by Prof Scribante's research group, published by you children, should also be emphasised.

10.3390/children8060432

Materials and methods: how was the sample size calculated?

Very confusing results, rearranging tables and rendering images and tables themselves in high resolutions.

Discussion: future goals are missing

Conclusions, rephrase them according to the changes

Bibliography: Add required bibliography.

Author Response

RESPONSE TO COMMENTS OF REVIEWER 2

Manuscript of considerable interest for the dental sector, it needs a major revision before it can be published.

Comment: Abstract: to better highlight the results obtained.

 RESPONSE: Thank you for this comment. The results are now highlighted.

Abstract Background: Supernumerary teeth refer to extra teeth that exceed the usual number in dentitions. Mesiodens is a particular form of supernumerary tooth which is located in the premaxilla region. The objective of the study was to investigate the genetic etiology of extra tooth phenotypes including mesiodens and isolated supernumerary teeth. Methods: Oral and radiographic examinations and whole exome sequencing were performed on every patient in our cohort of 122 patients, including 27 patients with isolated supernumerary teeth and 94 patients with mesiodens. A patient who had multiple supernumerary teeth also had odontomas. Results: We identified a novel (c.8498A>G;p.Asn2833Ser) and six recurrent (c.1603C>T;p.Arg535Cys, c.5852G>A;p.Arg1951His, c.6949A>T;p.Thr2317Ser; c.1549G>A;p.Val517Met,  c.1921A>G;p.Thr641Ala, and c.850G>C;p.Val284Leu) heterozygous missense variants in FREM2 in eight patients with extra tooth phenotypes. Conclusion: Biallelic variants in FREM2 are implicated in autosomal recessive Fraser syndrome with or without dental anomalies. Here we report for the first time that heterozygous carriers of FREM2 variants had phenotypes including oral exostoses, mesiodens and isolated supernumerary teeth.

Comment: Keywords: are they all present on MeSH?

 RESPONSE

Yes.

Comment: Introduction: all the other pre- and post-natal defects linked to genetic compositions, such as structural anomalies, as already studied by Prof Scribante's research group, published by you children, should also be emphasised.

10.3390/children8060432

RESPONSE

Thank you for this comment. I agree with you that the mutations in FREM2 might result in altered structural composition of the teeth. The authors suggested this valuable point for future study and cited the suggested reference.

Butera A, Maiorani C, Morandini A, Simonini M, Morittu S, Barbieri S, Bruni A, Sinesi A, Ricci M, Trombini J, Aina E, Piloni D, Fusaro B, Colnaghi A, Pepe E, Cimarossa R, Scribante A. Assessment of Genetical, Pre, Peri and Post Natal Risk Factors of Deciduous Molar Hypomineralization (DMH), Hypomineralized Second Primary Molar (HSPM) and Molar Incisor Hypomineralization (MIH): A Narrative Review Children (Basel). 2021 May 21;8(6):432. doi: 10.3390/children8060432.

Comment: Materials and methods: how was the sample size calculated?

RESPONSE: As mentioned in the Materials and Methods part. We collected as many cases as possible. We studied 122 patients with various kinds of isolated extra tooth phenotypes. To the best of our knowledge, we have studied the highest number of patients with extra tooth phenotypes. More than any studies in the literature. Statistically, more cases are always better.  Thank you for this comment.

2.2. Patients

Oral and radiographic examinations were performed on our cohort of 122 patients affected with various kinds of isolated extra tooth phenotypes, which included 94 patients with isolated mesiodens and 27 patients (10 females; 17 males) with isolated supernumerary teeth.

Comment: Very confusing results, rearranging tables and rendering images and tables themselves in high resolutions.

 RESPONSE: The Figures and Tables are now in high resolution.

Comment: Conclusions, rephrase them according to the changes

 RESPONSE: Thank you for this comment. It is rewritten as followed.

5. Conclusion

We report for the first time that heterozygous carriers of FREM2 may have phenotypes including oral exostoses, mesiodens and isolated supernumerary teeth.

Comment: Discussion: future goals are missing

Comment: Bibliography: Add required bibliography.

RESPONSE: Thank you for your great comments.  We added the section 4.9 Future studies in order to elaborate your valuable thoughts. The authors think that in the future the structural composition of the supernumerary teeth should be investigated. The paper by Butera et al 2021 is cited.

                        4.9 Future studies

Since FREM2 is a vital component of the FRAS1-FREM2-FREM1 complex and subsequent abnormal epithelial-mesenchymal interactions. Therefore, disruption of epithelial-mesenchymal interactions as a result of FREM2 mutations may have the effects on tooth development including tooth mineralization [Butera et al., 2021]. It is suggested that structural composition of the supernumerary teeth should be studied in the future.

We report heterozygous carriers of FREM2 in patients with extra tooth phenotypes. Bi-allelic variants of FREM2 are implicated in Fraser syndrome as previously mentioned [27,28]. The study of dental conditions of the heterozygous parents of the patients affected with Fraser syndrome has not been done. It is suggested that the dental evaluation be performed in the parents of patients with Fraser syndrome.

Reviewer 3 Report

Dear Authors,

thank you for your work submitted to the journal.

I found it interesting and really relevant from a scientific point of view. Genetic studies in dentistry are surely important. 

I kindly ask you to consider the following comments to imprve the quality of your manuscript:

- At the end of the introduction please specify the statistical null hypothesis

- In the discussion please specify wheter the statistical null hypothesis has been accepted or not

- I really appreciate the conclusion and the limitations section. I think it could be interesting to add a section related to future perspectives. In particular I think it could be interesting to discuss that it could be interesting to evaluate the morphological composition of suprannumerary teeth. you could refer for example to the following article:

https://onlinelibrary.wiley.com/doi/10.1111/odi.14388

Yours faithfully,

The Reviewer

Author Response

RESPONSE TO COMMENTS OF REVIEWER 3

Dear Authors,

Comment: Thank you for your work submitted to the journal. I found it interesting and really relevant from a scientific point of view. Genetic studies in dentistry are surely important. 

RESPONSE: Thank you for your kind word towards our work.

I kindly ask you to consider the following comments to improve the quality of your manuscript:

Comment: At the end of the introduction please specify the statistical null hypothesis.

Comment: In the discussion, please specify whether the statistical null hypothesis has been accepted or not

RESPONSE: Thank you for these comments. However, such statistical work-up is typically not done in genetic association studies, and that the authors would prefer not to add this to the manuscript, as it is not done in this area of genetics.

Comment: I really appreciate the conclusion and the limitations section. I think it could be interesting to add a section related to future perspectives. In particular I think it could be interesting to discuss that it could be interesting to evaluate the morphological composition of supernumerary teeth. you could refer for example to the following article:https://onlinelibrary.wiley.com/doi/10.1111/odi.14388

Butera A, Maiorani C, Morandini A, Simonini M, Morittu S, Barbieri S, Bruni A, Sinesi A, Ricci M, Trombini J, Aina E, Piloni D, Fusaro B, Colnaghi A, Pepe E, Cimarossa R, Scribante A. Assessment of Genetical, Pre, Peri and Post Natal Risk Factors of Deciduous Molar Hypomineralization (DMH), Hypomineralized Second Primary Molar (HSPM) and Molar Incisor Hypomineralization (MIH): A Narrative Review. Children (Basel). 2021 May 21;8(6):432. doi: 10.3390/children8060432.

RESPONSE: Thank you for your great comments.  We added the section 4.9 Future study in order to elaborate your valuable thoughts. The authors think that in the future the structural composition of the supernumerary teeth should be investigated. The paper by Butera et al is cited.

            4.9 Future studies

Since FREM2 protein is a vital component of the FRAS1-FREM2-FREM1 complex and subsequent abnormal epithelial-mesenchymal interactions. Therefore, disruption of epithelial-mesenchymal interactions as a result of FREM2 mutations may have the deleterious effects on tooth development including tooth mineralization [39]. It is suggested that structural composition of the supernumerary teeth should be studied in the future.

We report heterozygous carriers of FREM2 in patients with extra tooth phenotypes. Bi-allelic variants of FREM2 are implicated in Fraser syndrome as previously mentioned [27,28]. The study of dental conditions of the heterozygous parents of the patients affected with Fraser syndrome has not been done. It is suggested that dental evaluation be performed in the parents of patients with Fraser syndrome.

Yours faithfully,

The Reviewer

Round 2

Reviewer 2 Report

The manuscript has been properly revised, it can be published

Reviewer 3 Report

Dear Authors,

thank you for your improvements to the manuscript.

All my comments have been addressed.

The Reviewer